# Trends in Molecular Diagnosis of Nosocomial Pneumonia Classic PCR vs. Point-of-Care PCR: A Narrative Review

**DOI:** 10.3390/healthcare11091345

**Published:** 2023-05-07

**Authors:** Andrei-Mihai Bălan, Constantin Bodolea, Sebastian Daniel Trancă, Natalia Hagău

**Affiliations:** 1Department of Anaesthesia and Intensive Care 2, “Iuliu Hatieganu”, University of Medicine and Pharmacy Cluj-Napoca, 400012 Cluj-Napoca, Romaniahagaunatalia@gmail.com (N.H.); 2Department of Anaesthesia and Intensive Care, Municipal Clinical Hospital, 400139 Cluj-Napoca, Romania; 3Emergency Department, The Emergency County Hospital Cluj, 400347 Cluj-Napoca, Romania; 4Department of Anaesthesia and Intensive Care, “Regina Maria” Hospital, 400221 Cluj-Napoca, Romania

**Keywords:** pneumonia, point-of-care PCR, molecular diagnosis

## Abstract

Nosocomial pneumonia is one of the most frequent hospital-acquired infections. One of the types of nosocomial pneumonia is ventilator-associated pneumonia, which occurs in endotracheally intubated patients in intensive care units (ICU). Ventilator-associated pneumonia may be caused by multidrug-resistant pathogens, which increase the risk of complications due to the difficulty in treating them. Pneumonia is a respiratory disease that requires targeted antimicrobial treatment initiated as early as possible to have a good outcome. For the therapy to be as specific and started sooner, diagnostic methods have evolved rapidly, becoming quicker and simpler to perform. Polymerase chain reaction (PCR) is a rapid diagnostic technique with numerous advantages compared to classic plate culture-based techniques. Researchers continue to improve diagnostic methods; thus, the newest types of PCR can be performed at the bedside, in the ICU, so-called point of care testing—PCR (POC-PCR). The purpose of this review is to highlight the benefits and drawbacks of PCR-based techniques in managing nosocomial pneumonia.

## 1. Introduction

Pneumonia is a disease characterized by lower respiratory tract infection and alveoli inflammation. This condition is caused by viral, bacterial, or less frequent fungal infections. The treatment involves antiviral, antibacterial, or antifungal medications [1]. Pneumonia may also be a nosocomial infection: hospital-acquired pneumonia (HAP), caused by pathogens from the hospital environment which colonize the respiratory tract of patients admitted for longer than 48 h, and ventilator-associated pneumonia (VAP). VAP is the most frequent nosocomial infection in endotracheally intubated patients in intensive care units (ICU) (encountered in 5–40% of such cases) and usually occurs within 48–72 h from initiation of mechanical ventilation [2,3,4,5,6]. VAP is associated with longer hospital stays, ventilator dependence, higher medical assistance-related costs, infection and ventilation-associated complication rates, and mortality (up to 10% higher than in patients without VAP) [2,4,5,7,8,9].

In 2019, 498 million people globally suffered from respiratory tract illnesses (pneumonia, bronchiolitis). Among such cases, HAP varied from 5 to 10 cases per 1000 admissions, while VAP occurred in 10–25% of the patients [10,11,12]. Lower respiratory tract infections caused over 2.49 million deaths in 2019, from which HAP is considered the cause of death in 20–30% of cases, with VAP as the putative cause in 20–50% of them [10,12,13].

The diagnosis of VAP is established based on clinical suspicion, modified imagistic lung evaluation (classical X-ray or CT-scan), and positive microbiological cultures from the lower respiratory tract. Samples can be obtained non-invasively, as is the case for endotracheal aspiration, or invasively, by bronchoscopy, in the case of protected brush sampling (PBS) or bronchoalveolar lavage (BAL) [14].

Depending on the time of onset, VAP can be caused by susceptible to regular antibiotics microorganisms (*Staphylococcus aureus*, *Haemophilus influenzae*, *Streptococcus pneumoniae*), especially within the first four days from the initiation of mechanical ventilation (early-onset VAP), or by multidrug-resistant (MDR) bacteria (*Staphylococcus aureus*, *Pseudomonas aeruginosa*, *Acinetobacter*, *Enterobacter*, *Klebsiella*, *Serratia*), after more than five days since ventilation initiation (late-onset VAP) [3]. Microbial flora varies widely from one hospital to another and even within the same institution. Therefore, for antibiotic therapy to be as adapted as possible, the local ecology must always be considered [2,4,15].

MDR pathogens are problematic in respiratory infections and responsible for most severe cases and deaths. Treatment of MDR pneumonia is difficult and requires the most specific diagnosis as early as possible to provide adequate therapy in accordance with the pathogens’ sensitivity. Resistance to cephalosporins due to extended-spectrum β-lactamase expression is very concerning, as is the spread of carbapenemase-producing strains. For example, up to two-thirds of *Acinetobacter baumannii* strains leading to VAP show resistance to carbapenems [2,16,17,18,19].

Regarding pathogen identification, cell cultures may take at least 24–48 h for results, during which patients receive empiric broad-spectrum antibiotic therapy. The initiating treatment must consider MDR infection risk factors and the patient’s condition (presence of shock, degree of lung damage, comorbidities) [18,20,21]. The most frequent microorganisms involved in VAP are *Pseudomonas aeruginosa*, *Acinetobacter* spp., *Klebsiella pneumoniae*, *Enterobacter* spp., etc.; thus, therapy most often relies on the antibiotic combination to cover a broader spectrum of microorganisms, ensuring that treatment is not futile, and reduce the rates of mortality and complications associated with multidrug therapy [22]. Usually, when radiologic findings suggest lung infiltrates, accompanied by two or more clinical signs, antimicrobial treatment is started. Still, frequent use of nonspecific antibiotics leads to developing multidrug resistance and other drug-related complications, such as *Clostridioides difficile* enterocolitis. However, delaying VAP treatment increases mortality, so patients receive first-line antibiotics, with adjustments to therapy based on clinical response and microbiological testing results [2,23,24].

Early VAP diagnosis is essential for ensuring adequate antimicrobial treatment as soon as possible, reducing the risk of MDR organism emergence and complications induced by the pathogens or antibiotic treatment [3,25]. To start the most appropriate treatment for the patient with nosocomial pneumonia, we need to identify the infective pathogen as soon as possible and, more than that, to know the susceptibility of that pathogen to our arsenal of antibiotics or other drugs used to treat the infection. A very rapid and helpful way to have both simultaneously is by using PCR-based methods, especially considering that the gold standard for bacteria and antibiotic resistance identification needs more than 24–48 h [26].

The lung, once thought to be a sterile environment, has its specific bacterial population, even in healthy individuals. The first study which pointed this out was the one conducted by Hillty in 2010, which detected the lung microbiota using sequencing [27]. This supposition was confirmed by Erb-Downward et al., who investigated BAL specimens from healthy individuals but also from patients with lung diseases. They showed that even in healthy lungs, we have bacteria such as *Pseudomonas*, *Streptococcus*, *Haemophilus*, *Moraxela*, *Prevotella*, and *Fusobacteria* [28]. When different factors interfere with the normal functioning of the lungs, these species can uncontrollably multiply and can lead to pulmonary infections [29].

Over time, molecular diagnosis techniques have progressed significantly, and PCR becoming increasingly used. This technology detects bacterial DNA and is much quicker, more sensitive, and specific for identifying singular or multiple pathogens and genes involved in antimicrobial resistance [2,30].

The purpose of this review is to highlight the PCR-based methods used in managing nosocomial pneumonia and their benefits and also what needs to be taken into account to not under or over-treat the patients. In addition, this review aims to overview existing PCR techniques used for diagnosing pneumonia, particularly VAP. At the same time, we tried to draw a comparison between the classic and bedside (point-of-care-POC) PCRs.

To write this narrative review, we analyzed the most known databases (Pubmed Central, Google Scholar, Embase, Cochrane) and used search terms such as pneumonia, hospital-acquired infections, ventilator-acquired pneumonia, ventilation-associated pneumonia, polymerase chain reaction, point-of-care PCR, rapid diagnosis, and similarities. In reviewing these articles, we carefully tried to select them, looking at the most relevant results that concord with most other similar studies.

### 1.1. Polymerase Chain Reaction—Short Technical Overlook

PCR is a lab technique developed in 1980 by Kary Mullis to rapidly produce copies of specific DNA or RNA sequences [31]. PCR involves short fragments of lab-synthesized DNA, called primers, which will select a specific genome sequence for amplification. The classic PCR sequence has three main steps: denaturation, renaturation, and extension. Then, the three steps are repeated until sufficient amplification (approximately 25 cycles) of the sequence of interest [32,33,34].

Unlike the classic one, real-time PCR monitors the amount of DNA along the amplification cycles. It uses fluorescent probes, which emit light signals, to detect the DNA amount during each PCR cycle. Amplifying DNA through PCR is more efficient at the beginning of the reaction, and using real-time PCR allows measuring the result of this phase, called the exponential phase [34].

Although other methods for measuring nucleic acid quantities exist (PCR-ELISA, HPLC, electrophoretic analysis), these are plagued by numerous drawbacks: less sensitivity, more work-intensive and time-consuming, reliance on the use of radioactivity, and predisposition to cross-contamination (minimal in the case of PCR due to lack of post-PCR manipulation) [34,35,36]. Unlike the previously described methods, real-time PCR is extremely sensitive, which allows the detection of fewer than five copies of the targeted sequence. However, sometimes even a single one will suffice. Thus, real-time PCR allows the analysis of tiny samples, such as minuscule lysates and tissue fragments from biopsies [34]. Another advantage of this method is specificity—it can distinguish specific sequences from a complex DNA mixture and identify genetic material belonging to a particular pathogen in a given simple. Among the uses of real-time PCR, one can mention: detecting and identifying bacteria and viruses and determining gene expression or viral load. Pinpointing the pathogens responsible for various infections is vital for prescribing appropriate treatment as soon as possible, and PCR, to this end, is even faster than traditional plate culture techniques. Mayo Clinical Microbiology Lab has noticed a drop-in analysis time for six different pathogens from 1–14 days using traditional methods to 30–50 min using real-time PCR, with similar or even better sensitivity [34,35,37]. Microbial load is sometimes linked to disease severity, so microbiology labs use this method to document disease progression and treatment efficacy [29,34].

However, PCR has certain limitations, one of which is identifying genetic material belonging to dead and living pathogens, while cell cultures identify only viable microorganisms; this means that PCR requires further result interpretation [34]. The first challenge to molecular diagnosis in pneumonia is the risk of falsely incriminating non-infectious or colonizing microorganisms, leading to unnecessary treatment, as would be the case when including microorganisms with low pathogenicities, such as yeasts and *coagulase-negative staphylococci* [38,39,40,41]. However, quantitative PCR could offer information regarding pathogen density and distinguish between colonization and infection [38,42,43].

Real-time PCR is also susceptible to PCR alteration by compounds encountered in certain biological products. For example, clinical and forensic uses for real-time PCR are limited by the inhibition caused by products such as hemoglobin and urea [29].

Another major drawback of PCR was that it required DNA as a reading frame due to the impossibility of RNA amplification. This limitation was solved by using an enzyme called reverse transcriptase (RT), which generates complementary DNA (cDNA) from RNA frames, which can be used for various purposes, including PCR. Reverse transcriptases are enzymes used in nature by retroviruses, which include the human immunodeficiency virus and hepatitis B virus; the virus-generated cDNA can then be introduced into the host cell genome. When this enzyme is used, the process is called RT-PCR and is used to analyze RNA molecules; it has become the most popular method of quantifying mRNA levels [29,44].

PCR techniques continue progressing, and POC-PCR is an increasingly used method, especially for detecting infectious agents. Point-of-care testing is conducted in the patient’s vicinity without depending on a central laboratory, which significantly improves diagnostic methods. Developing these technologies for identifying infectious diseases is vital in countries with poor medical infrastructure since these conditions are responsible for over 50% of infant deaths [32,45,46,47,48]. These POC devices are used more and more frequently, and their employment in the United States is estimated to increase by 15% over the coming years [49]. Currently, the most widely utilized POC systems monitor hemoglobin concentration, thrombocyte function, test for HIV, group A streptococci, pregnancy, etc. [45,50,51]. As previously noted, POC-PCR is of great interest in diagnosing and treating pneumonia cases. POC-PCR devices are manufactured to integrate and automate all steps required for molecular analysis—extraction and purification of nucleic acids and detection based on PCR. Miniaturization should enable more frequent use of these devices [32,52,53]. They do not require trained and special medical personnel; the devices can be used with minimal training in the wards where the patients are treated [54].

In the case of various infections, such as VAP, quickly identifying the incriminated microorganism is needed to improve the patient’s outcome. Multiplex-PCR (mPCR) is one of the fastest solutions to this problem. It allows rapid, simultaneous detection and quantification of multiple pathogens or antibiotic resistance genes in various samples simultaneously [55,56,57,58,59,60,61].

### 1.2. Multiplex PCR—The Classic Method and Its Usefulness in Managing Pneumonia

As previously mentioned, some researchers consider that one of the main limits of mPCR is not being able to discriminate between living and non-viable pathogens. However, other authors view this as an advantage; some multiplex PCR tests correctly identify pathogens even if antimicrobial therapy has already been initiated [62,63].

Numerous studies have proven that PCR has higher sensitivity in detecting bacteria or viruses when compared to cell cultures [64,65,66]. Another study investigated patients’ results obtained through 112 cell cultures and 103 PCR tests; it, too, proved that PCR is far more sensitive than classic culturing, with 36 (32%) positive cultures and 55 (53%) positive PCR tests [55,67]. However, regarding their accuracy in diagnosis, it should be taken into account that all of these molecular-based technologies can not certainly differentiate a patient with colonization from one with infection, so they should be integrated into a constellation of criteria for diagnosis of pneumonia [68].

It seems that mPCR could be successfully used for blood samples since it has higher sensitivity than hemocultures. Hemocultures attempt to isolate and culture microorganisms in the patient’s blood [55]. Studies show that bacteremia is present in only 20% of all cases of pneumonia, lowering hemocultures’ sensitivity when attempting to diagnose such cases. When formulating a diagnosis based on blood samples, PCR seems more appropriate [55,69]. A study assessing the usefulness of testing for the *ply* gene in patients’ blood showed a positive result in 22 of 40 patients (55%) with pneumococcal pneumonia; meanwhile, hemocultures had a sensitivity of 28% (11 of 40 patients). PCR results were negative in all 30 patients with non-pneumococcal pneumonia, rendering it 100% specific [55].

Banerjee et al. discovered a significant drop in broad-spectrum antibiotic treatment duration for patients with pneumonia whose blood samples were analyzed by way of PCR vs. standard culture identification (44 vs. 56 h, *p* = 0.01), as well as improved narrow-spectrum antimicrobial drug use, without any significant differences in terms of mortality, length of stay, or cost [62,70].

Baudel et al. investigated the accuracy of multiplex PCR in diagnosing VAP and hospital-acquired pneumonia (HAP) in ICU patients [71]. They used the LightCycler 2.0 SeptiFast (Roche Diagnostics, Mannheim, Germany) kit, meant for use with blood samples. Of the 65 patients with pneumonia suspicion and 53 with confirmed pneumonia, the test detected a pathogen in 66% of the samples. The identification rate was up to 82% after accounting for samples with microorganisms for which no probe existed in the system. In this study, prior antibiotic therapy did not influence the detection rate [62,72].

VAPChip (Eppendorf Array Technologies, Namur, Belgium) is a system comprising a plastic cartridge that combines multiplex PCR and hybridization from probes obtained by endotracheal aspiration, BAL, or PBS. This diagnostic test’s main advantage is its ability to simultaneously identify multiple species and resistance genes. [73].

The main advantages and drawbacks of using PCR in pneumonia diagnosis are presented in Table 1.

### 1.3. POC-PCR—A Game Changer?

Using POC-PCR presents numerous advantages, some of which will be presented next. The most obvious advantage of these analysis systems is time-saving, which is extremely important for timely decision-making regarding adequate treatment and rapid initiation [75,76,77]. Samples analyzed with POC systems require no transportation and processing from the drawing point to the central analysis lab, shortening the waiting time until obtaining a result by at least 1–2 h. This allows some tests to be run quickly on demand from or by medical personnel from a department without depending on the transportation of samples to other medical institutions or labs [77,78]. Classical PCR analysis of nucleic acids involves multiple manual steps, such as sample dilution or mixing, which requires extra work time, as well as specialized personnel and devices; all these means that PCR can only be performed in central labs, POC-PCR does not need all these steps, they are automated. POC-PCR is far better in terms of time than traditional microbiological methods, which rely on plate cultures and take longer than 48 h to provide a result [79,80,81,82]. A study by Kunze et al. supports that multiplex POC-PCR requires far less time than cultures. In this study, the average time until a result for cultures was 71.2 h, whereas only 6.5 h was necessary for a multiplex POC-PCR test to provide a result. However, cell cultures’ results were 100% valid, whereas only 60% of POC-PCR tests were valid [83].

A randomized control study investigated whether automated rapid PCR (rPCR), which detects methicillin-resistant *S. aureus* (MRSA) in BAL, could safely reduce vancomycin or linezolid consumption for VAP cases caused by this microorganism [84,85]. A total of 22 patients received antibiotics based on the rPCR, and 23 others were subjected to routine care, which involved empirical treatment. The duration of anti-MRSA treatment was significantly shorter in the intervention group (32 vs. 72 h). Moreover, in-hospital mortality was reported as 14% in the intervention and 39% in the control group [85].

Early specific treatment reduces the risk of developing broad-spectrum antibiotic resistance, earlier patient recovery and discharge, and reduced medication adverse effects rates. Although POC-PCR is sometimes more expensive than traditional diagnostic methods or classic PCR, the shorter hospitalization and the reduced rate of inadequate treatment may compensate for the cost difference [32,45,86].

In a study by Perez et al., rapid identification of the incriminated pathogen was associated with a drop in average hospital stay from 11.9 to 9.3 days, a 20% total stay improvement. Moreover, patients tested with these rapid methods received antiviral medication one and a half days earlier than the others, which is extremely important when referring to certain antiviral drugs, whose maximum effectiveness is achieved only by the administration within 48 h from symptom onset [45,71,87].

Most POC tests would previously detect antigens specific to influenza and syncytial respiratory viruses. However, new methods of multiplex POC-PCR were developed, which, using fresh samples, aim to detect a group of microorganisms responsible for respiratory infections, including pneumonia [86,88,89,90].

An important aspect of using PCR tests is that they allow diagnosing patients with viral pathogens that only require antiviral treatment, despite clinical suspicion of co-infection. In the case of bacteria detection, it is important that PCR-based molecular tests, such as BioFire Film Array or Curetis Unyvero, allow the detection of non-core pathogenic agents, such as MRSA (mecA and MecC genes), MDR, and non-MDR *Pseudomonas*, *Enterobacterales* producing ESBL and resistant to carbapenems and *Acinetobacter.* A disadvantage of these tests is that some do not differentiate colonization from infection [91].

One of these tests is the pneumonia cartridge in the mPCR Unyvero system (Curetis GmbH, Holzgerlingen, Germany), which can identify 20 bacteria and one fungus often associated with VAP, as well as 19 of their resistance markers [30,83]. This pneumonia panel from Unyvero has correctly identified the pathogens responsible for VAP in 73% of patients and the respective resistance mechanism in 67%, with differences from classic microbiologic methods attributed to the incriminated pathogen, *P. aeruginosa* having the highest discordance rate [92].

Luyt et al. included 93 samples from patients suspected of VAP which were analyzed by conventional cultures and the Unyvero HPN cartridge. The Unyvero cartridge correctly identified pathogens in 68 samples, and 25 of these had different results compared to the culture method. Among the discordant samples, there were eight samples for which the pathogen was not included in the Unyvero cartridge panel. According to the study, the Unyvero cartridge did not show the best diagnostic performance compared to conventional microbiological methods. The results obtained by Luyt et al. suggested that the limitations of these tests are overdetection or underdetection. The overdetection is due to identifying non-viable pathogens or colonizing bacteria, but the more recent cartridges—Unyvero P55 or Unyvero HPN used by Gadsby et al., can quantify the bacterial load and give a positive result only when this load is sufficiently high [92]. One more Unyvero HPN cartridge test was evaluated by comparison with conventional microbiological methods. The test identified 104 bacteria compared to the culture method, which identified 128 bacteria. The HPN test presented a sensitivity of 80% and a specificity of 99%, a positive predictive value of 87%, and a negative predictive value of 99%. It concluded that 20 of the 95 patients could have received a suitable antibiotic sooner, 37 patients could have benefited from early de-escalation, and for three patients, the treatment could have been optimized. Furthermore, out of 17 patients treated with extended-spectrum antibiotics, 10 could have benefited from de-escalation in the following hours [93].

Similar to the Unyvero system, the BioFire FilmArray (BioFire Diagnostics, Salt Lake City, UT, USA) respiratory panel detects 14 respiratory viruses (human influenza virus A and B, adenovirus, coronaviruses HKU1, NL63, 229E, and OC43, human metapneumovirus, human rhino- and enterovirus, parainfluenza virus types 1–4, syncytial respiratory virus), three bacteria (*Mycoplasma pneumonia*, *Chlamydia pneumonia*, *and Bordetella pertussis*), as well as fungi. This panel is a fully automated multiplex PCR system requiring only 2 min of practical work and provides results in approximately one hour, with a specificity of 99% and a sensitivity of 81% [86,94]. The FilmArray respiratory panel is the first FDA-approved test for qualitatively detecting nucleic acid targets in nasopharyngeal swab samples [95,96]. Some studies claim that the FilmArray RP can detect up to 17 viral targets, not just 14, and the three bacterial species most commonly associated with community-acquired pneumonia [95,97]. Recently, FilmArray RP was used to prove that over 24% of HAP episodes were related to viral infection alone or viral and bacterial co-infection [95,98]. This information could have important implications for modifying or de-escalating antibiotic therapy [95,97].

Another respiratory dedicated multiplex POC-PCR, the BioFire^®^ FilmArray^®^ Pneumonia plus Panel (FilmArray PP), enables rapid and accurate automated testing for 27 bacteria and viruses that cause pneumonia and other lower respiratory tract infections (LRTI), as well as for seven genetic markers of antibiotic resistance. Based on a study that included 112 samples from patients with COVID-19, it was evaluated how PCR testing influenced the approach to antibiotic therapy in these patients. The samples were obtained from 67 patients suspected of community-acquired pneumonia (CAP), HAP, or VAP. The study’s results showed that the suspicions of pneumonia were true in 13% of cases for CAP, 13% for HAP, and 39% of cases for VAP. The sensitivity of the BioFire^®^ FilmArray^®^ Pneumonia plus Panel test for these samples was divided to be 83%, a specificity of 99.1%, a negative predictive value of 99.9%, and a positive predictive value of 52.1%. When these values were recalculated only for the microorganisms included in the test system, the test’s sensitivity was 100%, the specificity 98.8%, the negative predictive value 100%, and the positive predictive value 52.1%. PCR testing determined the antibiotic treatment modification for 34% of the patients [99].

FilmArray PP can also distinguish clinically relevant pathogens from colonizing bacteria or normal flora. These tests provide a result based on the abundance of nucleic acid. The study by Lee et al. analyzed 59 endotracheal aspirates and BAL samples and detected at least one pathogen in 33 samples, obtaining a positivity rate of 55.9%. Among the positive samples, 14 cases of co-infections were identified (42.4%), the highest number of pathogens identified in a sample being 8. 27.1% of the samples, i.e., 16 out of 59, showed infection with viruses, the influenza A virus being the most common. Six samples showed infection with viruses and bacteria. Unlike the conventional method, the FilmArray platform identified 16 viral targets in 100% agreement with classic PCR testing. The results of this study show that FilmArray PP could change the prescription of antibiotics in 40.7% of patients [100].

Another study that considered the BioFire FilmArray panel evaluated its ability to detect the pathogens responsible for VAP. The FilmArray Blood Culture Identification (BCID) panel was used, proving high sensitivity for blood cultures [101,102,103,104,105]. There were 167 samples used which were analyzed by MS MALDI-TOF and also by BCID. The main organisms identified by both diagnostic tests were *Pseudomonas aeruginosa*, *Acinetobacter baumannii*, and *K. pneumoniaee*. Considering the potentially carbapenem-resistant species only as a group (*A. baumannii*, *P. aeruginosa*, and *K. pneumoniae*), the values were 78.6%, 98.1%, 87.3%, and 96.6% for sensitivity, specificity, PPV, and NPV, respectively. In addition, it has been reported that 111 Gram-negative isolates tested for the presence of the *KPC* gene. Its presence was identified in 5 *K. pneumoniae* isolates by conventional PCR, all of which were also detected by the BCID panel. The BCID panel also detected the *KPC* gene in two additional *K. pneumoniae* isolates. The authors believe that obtaining the results faster with the help of the BCID panel would have allowed 72 out of 150 patients (42%) to receive more appropriate therapy [105].

Data regarding resistance in pathogens collected using these tests were recently published in a sub-study of the PROGRESS study—a randomized, multicentric prospective study. It included 56 patients with community-acquired pneumonia and without risk factors for MDR pathogens and 34 patients with risk factors. Pneumonia testing had a detection rate of 72%, while conventional microbiologic testing had a rate of 10% (*p* < 0.001). These results support the value of this diagnostic test and its potential implementation in the clinical practice [85,106].

A new panel, Luminex NxTAG Respiratory Pathogen Panel (NxTAG-RPP, Austin, TX, USA), was introduced as a high-performance system, capable of detecting nucleic acids belonging to 21 respiratory viruses, including all pathogens detected by FilmArray, except for *B. pertussis*, but including *Legionella pneumophila* and the human bocavirus [95,107]. Comparing these two panels showed complete matching in 98.8% (318/322) of positive results [95,108].

Cepheid’s GeneXpert device (Cepheid, Sunnyvale, CA, USA) is another point-of-care pathogen detection system that relies on multiplex PCR technology, using an automated and integrated sample preparation, nucleic acid extraction, and target sequence amplification and detection system. These devices have a specificity of 100% and 97–100% sensitivity, capable of identifying pathogens from nasal aspirate, nasopharyngeal swabs, nasal or nasopharyngeal lavage, and blood samples. Besides the very little time required, another advantage is the cost of just EUR 40 per sample [55]. This system can trace *MRSA* from an isolated colony in less than an hour. In the original study by Huletsky et al., a real-time PCR test was developed for the first time to target the DNA sequences in the orfX region where the staphylococcal cassette chromosome mec (SCCmec) integrates into the *S. aureus* chromosome and in 2007, another real-time PCR MRSA test became available, which also targets DNA sequences in the orfX-SCCmec chromosome junction [109,110].

In Table 2, we tried to summarize the types of tests discussed and their most important characteristics.

The PCR-based MRSA/SA GeneExpert diagnosis platform was also studied at the Veterans Affairs Medical Center in Houston, proving that, for *MSSA* bacteriemia, the average time to adequate therapy initiation was reduced from 49.8 to 5.2 h, while the duration of unnecessary MSSA treatment was reduced by 61 h per patient [89,110].

Another study assessing the MRSA/SA SSTI Cepheid Xpert test’s performance on 135 lower respiratory tract secretion samples concluded that it had a sensitivity of 99% and a specificity of 72.2% [95,117]. These relevant data support implementing higher-quality methods of diagnosing VAP [118].

Although POC-based technologies keep progressing faster, products integrating such methods have limited availability owing to certain challenges which arise during development: providing reagents with extended stability, architectural integration of multiple systems (thermal, microfluid, and optic), identifying materials both affordable and biocompatible [32,119].

### 1.4. Finding AMR Genes—Another Helping Tool in Antibiotic Guidance

Another important aspect of using PCR technologies in pneumonia management and a way of avoiding antibiotic overtreatment and drug resistance is fast molecular pathogen and antimicrobial resistance (AMR) gene identification. Although the potential of rapidly identifying resistance mechanisms is continuously improving, molecular AMR gene detection is still imperfect, and using them could not avoid overtreatment [38,40,76,120,121]. In a prospective observational study including patients with suspected VAP or HAP, multiplex PCR could perfect empirical antimicrobial therapy and reduce broad-spectrum antibiotic use if clinically applied [38,93]. Identifying the particular microorganism that carries a certain resistance gene is difficult, so a resistance mechanism might be present in an irrelevant or colonizing germ [38,40,122,123,124].

VAPChip system, mentioned before, can identify 24 genes associated with antibiotic resistance. Bogaerts et al. showed in a study on 292 isolated samples that VAPChip correctly identified 289 of them (99%). One of the three erroneously identified samples contained a mutant gene, while two lacked it entirely. The resistance genes were pinpointed with a sensitivity of 98.7% and a specificity of 97.7%, correctly detecting 269 of 272 representatives of the three tested β-lactam resistance gene families (blaTEM, blaSHV, blaCTX-M) [73].

Similarly, some other platforms can detect AMR genes. Unyvero can detect 19 genes associated with antibiotic resistance, and FilmArray can detect seven, both of which proved a high sensibility and specificity among studies [73]. FilmArray Pneumonia Plus Panel provided a positive and negative percent agreement (PPA and NPA) of 80–100% and 91.4–100%. It also detected 15 pneumonia pathogens responsible for pneumonia with a sensitivity rate of over 95% [76]. In another study, among the respiratory samples tested, four extended-spectrum β-lactamase (ESBL) blaCTX-M and three carbapenemase genes (blaIMP, blaNDM, and blaVIM) were detected. The detection efficiency of pathogens was 90% for positive samples and 97.4% for negative ones [103].

In a study, Unyvero’s HPN cartridge detected five of eight extended-spectrum beta-lactamase și 4 carbapenemases, while in another one, the P55 cartridge detected 18 patients with relevant AMR gene while the routine culture detected only 10 [76,125]. Unyvero’s system detects more AMR genes compared to FilmArray, including *bla* _TEM_ and *bla* _SHV,_ which codifies TEM respective SHV β-lactamases [125].

Another system that detects AMR genes is the aforementioned, GeneXpert. GeneXpert kit Carba-R (Cepheid, Sunnyvale, CA, USA) is capable of detecting carbapenemases such as KPC, oxacillinase-type carbapenemase (OXA-48, OXA-181, OXA-232), and metallo-beta-lactamases (MBLs) which include imipenemase MBL-1, New Delhi MBL, and Verona integron-encoded MBL) in about an hour (48 min) [126]. When comparing only the performance of the device’s AMR gene-detecting capability, the sensitivity and specificity rates were significantly higher (100% and 94.2%, respectively) [127].

Using this type of test panel is limited by its small number of primers to detect genes, and this can be improved by adding more important AMR genes to be detected [76].

### 1.5. Classic PCR—POC PCR Comparison

Although few studies have compared the two methods, we describe the main articles and their results in the following paragraphs, summarizing the main differences in Table 3.

The FilmArray Respiratory Panel (BioFire) platform is among the most widely used molecular testing methods at the patient’s point of care. As mentioned, it can simultaneously detect several pathogens and provide results in approximately 1 h. The study by Shengchen et al. showed that the sensitivity and specificity of this test varied from 92.3% to 97.9%, respectively, from 96.1% to 99.1% [94,128,129]. The study evaluated 800 patients between 16 October 2017, and 13 July 2018, and wanted to highlight how the use of POC-PCR influences the treatment of lower respiratory tract infections. The study results showed that the duration of treatment with intravenous antibiotics was shortened for patients tested with FilmArray from 8 days to 7 days (6–11 days of antibiotics for patients tested with PCR and 5–9 days of antibiotics for patients tested with POC-PCR). In addition, following the first results of the POCT test, eight patients stopped using intravenous antibiotics on the same day. Moreover, the average duration of hospitalization in the intervention group was shortened by approximately one day, from 9 days of hospitalization in the control group (7–12 days) to 8 days in the intervention group (7–11 days). The same authors found that if the FilmArray tests do not exceed USD 360, the average cost allocated to antibiotic treatment and hospitalization is lower for POCT-tested patients [128].

A study comparing the Unyvero P55 diagnostic test (Curetis AG) with bacterial culture methods and multiplex PCR tests showed that the test had a sensitivity of 56.9% and a specificity of 58.5%. The study included 74 samples obtained over 33 months from patients admitted to the intensive care unit of a hospital in Scotland. Although the sensitivity and specificity of the test did not reach the desired standards, the Unyvero P55 cart detected additional pathogenic organisms compared to the other diagnostic methods in 16.2% of the samples. It was also discovered that using the Unyvero P55 diagnostic test, the change in antibiotic treatment could have occurred in 60.7% of the patients who required a change in medication [130].

Besides savings due to faster discharge and less wasteful drug administration, POC-PCR does not require additional qualified personnel. Therefore, nucleic acid analysis with these methods can be carried out by personnel with minimal training, although the advantages of such tests can be influenced by the work technique, especially by delaying testing, which inherently leads to delaying appropriate treatment [32,86,131].

Another advantage of POC-PCR is the need for fewer samples, making them less invasive. Not needing to be performed from blood samples is an advantage for the patient [132,133]. There are cases, such as children, the elderly, and oncologic or ICU patients, in which drawing large amounts of blood is extremely difficult, which makes POC-PCR a safe alternative for these situations. Just as advantageous is using POC-PCR on neonatology units to reduce bleeding risks [45,134,135]. In classic PCR, sample preparation is performed before sequence amplification, and this stage is extremely important since the purity and integrity of the nucleic acids could be affected; the quality of the genetic material influences the significance and reproducibility of the results. POC-PCR reduces the risk of damaging nucleic acids or amplicon contamination and prevents false positive results [32,136].

Studies have also compared GeneXpert to other rapid detection systems, such as BD GeneOhm (BD Diagnostics GeneOhm, Québec, Canada), since they both combine pathogen identification with antimicrobial resistance detection. GeneXpert relies on real-time PCR, whereas BD GeneOhm uses real-time PCR combined with a fluorescent hybridization probe for target molecules [109,118]. These tests can rapidly identify (1–2 h) specific genes for *S. aureus*, whether methicillin-resistant or not, from colonies, hemocultures, or samples from nasal secretions, skin, or soft tissues [113,118,137,138,139]. The sensitivity and specificity of these systems were 75 and 94.5%, respectively, for GeneXpert and 85.2 and 96.5%, respectively, for BD GeneOhm [113,118]. Other figures suggest up to 92% sensitivity for GeneXpert and 97.2% for the other, with similar specificities [118,140,141]. Two further studies assessed the systems’ efficacy in sputum samples [118,141]. In one of them, BD GeneOhm had high sensitivity (97%) and specificity (92%) when testing 32 positive cultures. In contrast, GeneXpert had 98.4% sensitivity and 79.4% specificity when applied to 71 positive quantitative cultures from suspected VAP cases; when qualitative cultures were considered, the specificity grew to 95.2% [118,141]. A separate experimental study investigated S. aureus in endotracheal aspirate samples from mechanically ventilated patients, comparing GeneXpert to two other PCR-based and three cell culture methods. The result was that the GeneXpert scored 100% in both sensitivity and specificity [85].

The main advantages of POC-PCR over classic PCR are shown in Table 3.

**Table 3 healthcare-11-01345-t003:** Advantages of POC-PCR over classic PCR.

Classic PCR	POC-PCR
Specialized medical personnel needed for the process	No need for qualified medical personnel
Medium sample volume needed	Very little sample volume necessary
Requires sample transportation from bedside to laboratory	Does not require sample transport, is performed near the patient
Requires extensive processing before amplifying	Only requires sample lysis before amplifying
Risk of nucleic acid degradation or contamination during processing	Minimal risk of degradation or contamination due to no processing required
Time until results—approximately 6 h	Time until results—approximately 3–4 h

The left column shows the characteristics of classic PCR, and the right column shows the characteristics of POC-PCR [27,72,78,87,124,128,134,135].

### 1.6. Knowledge Gaps and Feature Directions

As mentioned before, one of the most cited problems of these devices is the lack of clear differentiation between colonization and infection. This issue was partly addressed by the main two devices (Unyvero and BioFire) in giving semiquantitative results which can be correlated to the bacterial/viral load and interpreted in context, but a clear limit does not yet exist. Another aspect, yet difficult to solve, is that these devices do not comprehend all of the potential infecting pathogens and the AMR genes. For example, *S. maltophilia* is lacking from FilmArray, and *Citrobacter koseri* and *Raoultella* spp. are absent in both devices. Regarding the aspect of not searching for AMR genes, we believe that the main issue is the lack of detecting the carbapenemases presented by *Klebsiella* spp. (the devices search for the main CTX-M types of carbapenemases and others for Unyvero, which can cover at most 70% of *Klebsiella’s* carbapenemases). In addition, the device cannot differentiate between the AMR genes specific to the commensals and those specific to the pathogens [77,125].

Thankfully, there were made many advances in rapid molecular testing techniques for pathogen confirmation, especially in the field of making them easier and easier to use and faster to obtain a result. Taking into account their limitations, especially the one regarding distinguishing between colonist and pathogen and also the one regarding its costs, we believe that in order to include this type of diagnostic tools in international guidelines, researchers should focus on clearly directed studies regarding delimitation between colonization and infection and also cost-efficiency evaluation. Another problem that should be addressed in future studies is the device’s limitation in searching for all the pathogens that can be responsible for HAP or VAP—maybe the developers could extend the range of primers in order to fill this gap. To solve the differentiation issues, maybe in the future, there could be a mixture of technologies between metagenomic sequencing and mPCR [77].

## 2. Conclusions

Due to increased multidrug-resistant pathogens, early diagnosis of hospital and particularly ventilator-associated lung infections is vital to adequately treat patients from the onset, without the requirement for broad-spectrum antibiotics, which might not be effective against the causal agent. Until recently, the fastest and most specific diagnostic tool was based on the PCR technique. To analyze samples simply and rapidly, without the need for transport or intermediate preparatory steps, POC-PCR is the most appropriate method to identify VAP-causing pathogens, as well as any genes involved in antimicrobial resistance.

The main limitation of this review article is the lack of an important number of studies regarding this new method of microorganisms and gene identification. On the one hand, this can be explained by the cost that these devices have but also by the lack of including them in guidelines. There is also a lack of studies regarding cost-benefit analysis and also a comparison between the most used POC-PCR devices.

However, to reach a conclusion regarding the utility of such techniques in pneumonia management and, even further, to include them in management guidelines, larger, preferably randomized blind studies are necessary.

## Figures and Tables

**Table 1 healthcare-11-01345-t001:** Pros and cons of PCR.

PROS	CONS
High sensitivity	High costs when compared to traditional methods
High specificity—can distinguish a particular sequence from a complex mixture of DNA molecules	Some pathogens are not included in the multiplex panel
Extremely fast when compared to traditional methods	Can not assess susceptibility to all antibiotics—antibiogram remains gold standard
Minimal risk of cross-contamination	Non-viable organism detection complicates diagnosis between current and prior infection
Easy to do	
Can use blood as fluid sample, higher sensitivity than hemoculture	
Simultaneous identification of multiple microorganisms and resistance genes	
Antimicrobial therapy does not influence results	
Lower risk of multidrug-resistant pathogen emergence	
Fewer needlessly administered drugs	

The left column contains the advantages of using PCR in pneumonia diagnosis, and the right column presents the disadvantages of using PCR in pneumonia diagnosis [34,35,62,64,65,70,74].

**Table 2 healthcare-11-01345-t002:** Types of tests based on PCR technology and their characteristics.

Platforms	Pathogens	Sensibility	Specificity	Time
GeneXpert [55,76]	MRSA, MSSA, Influenza A/B, RSV, TB, and genes associated with antibiotic resistance (carbapenemases)	75–99%	72–100%	1 h
Curetis Unyvero system [30,55,76,77,94]	20 types of bacterial and fungal pathogens and 19 genes associated with antibiotic resistance	81%	99%	4 h
FilmArray Respiratory Panel [21,55,76,77,111,112]	22 types of pathogens: 18 types of viral pathogens and 4 types of bacterial pathogens	81–100%	98–100%	45 min
FilmArray Pneumonia Panel [21,76,100]	33 types of pathogens: 18 types of bacterial pathogens, 8 types of viral pathogens and 7 antimicrobial resistance genes	96.2%	97.2–98.3%	1 h
BD GeneOhm MRSA [113]	MRSA	85.2–97.2%	92–96.5%	1 h
VAPChip [4,88]	13 types of VAP-causing pathogens and 24 genes associated with antibiotic resistance	98.7%	97.7%	5 h
Luminex NxTAG Respiratory Pathogen Panel (NxTAG-RPP, Austin, TX) [111,114,115,116].	21 types of viral pathogens	80–100%	98.8–100%	1 h

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
