# Peer review of "Trends in Molecular Diagnosis of Nosocomial Pneumonia Classic PCR vs. Point-of-Care PCR: A Narrative Review"

_healthcare, 2023, doi:10.3390/healthcare11091345_

Round 1

Reviewer 1 Report (Previous Reviewer 2)

Recommendations incorporated.

Author Response

  • We modified the abbreviations and also re-checked the whole manuscript for others like this.
  • We also modified the titles of the tables as instructed and moved the information that you mentioned in the legends of the tables
  • We fixed table 3 editing issue and hope that is now fine. This was a problem that wasn’t obvious to us just after we printed and checked to see what did you try to point out.
  • We introduced a new section in the manuscript to point out how important finding these AMR genes is for a proper diagnosis and for specific antibiotic coverage.
  • We introduced a new section to cover what you have asked for. We tried to be as specific as possible and to cover the main aspects referring to our subject.
  • We took out from the article all of the citations which were older than 2005, also we covered new studies which are now cited in the manuscript

Reviewer 2 Report (New Reviewer)

I have read the review with high interest. However, I have the following comments that should be addressed:

1. Line 40,41: You have already used HAP and VAP abbreviations in the previous paragraph. Also, line 96: PCR abbreviation is mentioned, so no need to re-write them in full again.

2. Table 2: part of the title, including the citations, should be moved to the table legend not the title.

3. The line numbers are within table 3 making it very hard to read.

4. Table 4: similar point mentioned for table 2.

5. I highly recommend a clear separate section about using PCR to detect antibiotics resistant genes, not only to detect the presence of the microorganisms.

6. A more clear section about the knowledge gaps and future directions is recommended.

7. Although the authors cited a large number of references, many are very old, even 10-20 years ago. The authors are encouraged to consider the following, not limited to (addition or replace):

Alsayed, A.R., Abed, A., Khader, H.A., Al-Shdifat, L.M., Hasoun, L., Al-Rshaidat, M.M., Alkhatib, M. and Zihlif, M., 2023. Molecular Accounting and Profiling of Human Respiratory Microbial Communities: Toward Precision Medicine by Targeting the Respiratory Microbiome for Disease Diagnosis and Treatment. International Journal of Molecular Sciences, 24(4), p.4086.   Dhesi, Z., Enne, V.I., O ‘Grady, J., Gant, V. and Livermore, D.M., 2020. Rapid and point-of-care testing in respiratory tract infections: an antibiotic guardian?. ACS Pharmacology & Translational Science, 3(3), pp.401-417.     Xu, E., Pérez-Torres, D., Fragkou, P.C., Zahar, J.R. and Koulenti, D., 2021. Nosocomial pneumonia in the era of multidrug-resistance: Updates in diagnosis and management. Microorganisms, 9(3), p.534.

Author Response

  1. Line 40,41: You have already used HAP and VA abbreviations in the previous paragraph. Also, line 96: PC abbreviation is mentioned, so no need to re-write them in full again.
  • We modified these abbreviations, and also re-checked the whole manuscript for others like this.

  1. Table 2: part of the title, including the citations, should be moved to the table legend not the title. And Table 4: similar point mentioned for table 2.
  • We also modified the titles of the tables as instructed and moved the information that you mentioned in the legends of the tables

  1. The line numbers are within table 3 making it very hard to read.
  • We fixed this editing issue and hope that is now fine. This was a problem that wasn’t obvious to us just after we printed and checked to see what did you try to point out.

  1. I highly recommend a clear separate section about using PC to detect antibiotics resistant genes. not oniy to detect the presence or the microorganisms
  • We introduced a new section in the manuscript to point out how important finding these AMR genes is for a proper diagnosis and for specific antibiotic coverage.

  1. A more clear section about the knowledge gaps and future directions is recommended
  • We introduced a new section to cover what you have asked for. We tried to be as specific as possible and to cover the main aspects referring to our subject.
  1. Although the authors cited a large number of references. many are ver old. even 10-20 vears ago. The authors are encouraged to consider the following, not limited to (addition or replace):
  • We took out from the article all of the citations which were older than 2005, also we covered new studies which are now cited in the manuscript. We also took your advice and used the information from the articles suggested by you.

Round 2

Reviewer 2 Report (New Reviewer)

All comments are now addressed and the paper is improved

This manuscript is a resubmission of an earlier submission. The following is a list of the peer review reports and author responses from that submission.

Round 1

Reviewer 1 Report

This review summarises some available reports on PCR for the diagnosis of VAP.

The authors might consider the following comment for improvement

1-Line 35 , please update the references  ,[ ….13] it should be 7 not 13

2-The background information in the introduction could be improved.  I would suggest to authors show how useful is PCR for the VAP and the rationale for performing this review  

3-Line 211 :  ……. have proven that PCR has higher sensitivity when compared to

cell cultures [65, 66].

This sentence should be revised and provide more accuracy. Not always that PCR has higher sensitivity as compared to culture. It depends on what you are looking for( diagnostic or infecting pathogens identification? DNA presence? ) It should be mentioned clearly 

Author Response

Reviewer nr. 1 

In the resubmitted manuscript, the modifications suggested for this reviewer are marked with the color red.

Comments and Suggestions for Authors

This review summarises some available reports on PCR for the diagnosis of VAP.

The authors might consider the following comment for improvement :  

1-Line 35 , please update the references  ,[ ….13] it should be 7 not 13

  • We changed the references and put them in ascending order from the beginning. In addition, we used Zotero software to ensure that we respect the order of the references.

2-The background information in the introduction could be improved.  I would suggest to authors show how useful PCR is for the VAP and the rationale for performing this review  

  • We wrote a few lines about the usefulness of the methods in diagnosing VAP, and also we added the purpose of the review in the introduction but also in the abstract (as requested by another reviewer)

3-Line 211 :  ……. have proven that PCR has higher sensitivity when compared to 

cell cultures [65, 66]. 

This sentence should be revised and provide more accuracy. Not always that PCR has higher sensitivity as compared to culture. It depends on what you are looking for( diagnostic or infecting pathogens identification? DNA presence? ) It should be mentioned clearly 

  • We re-formulated this phrase and tried to emphasize the drawbacks and limitations of the methods when diagnosing pneumonia.

Reviewer 2 Report

Please see the below comments and suggestions for the authors:

1.  Recommend adding purpose of review to the abstract

2.  Recommend omitting or re-writing second sentence on line 14. 

3.  Would add limited methods section.

4.  Would add limitations section to address limitations of type of review as well as address existing knowledge gaps and future research needs,

5.  Recommend address English style as appears there are multiple writers with various prose styles.  Re-write should be more consistent and address multiple incomplete sentences.  Recommend usage of commonly identified terms throughout the review.  Example:  length of stay vs duration, cost vs price, pathogen/microorganisms vs germs.

6.   Accuracy of PCR is mentioned but not much literature reviewed.  Would add for completeness.

7.  Would recommend adding morbidity, mortality data.

8.  A chart listing all the types of tests discussed and their characteristics would be summative.

Author Response

Reviewer nr. 2

In the resubmitted manuscript, the modifications suggested for this reviewer are marked with the color blue.

Comments and Suggestions for Authors

Please see the below comments and suggestions for the authors:

  1. Recommend adding purpose of review to the abstract

- We added a purpose in the abstract but also the introduction.

  1. Recommend omitting or re-writing second sentence on line 14. 

- We omitted this sentence.

  1. Would add limited methods section.

- We wrote six lines about the methods used to find articles for this review.

  1. Would add limitations section to address limitations of type of review as well as address existing knowledge gaps and future research needs,

- We added a section to address the article’s limitation in the conclusion part of the review.

  1. Recommend address English style as appears there are multiple writers with various prose styles.  Re-write should be more consistent and address multiple incomplete sentences.  Recommend usage of commonly identified terms throughout the review.  Example:  length of stay vs duration, cost vs price, pathogen/microorganisms vs germs.

- We modified all the different terms in the manuscript and tried to keep uniformity when referring to the same thing.

  1.  Accuracy of PCR is mentioned but not much literature reviewed.  Would add for completeness.

- We added new studies to our review, they are marked with the color green because another reviewer also recommended this.

  1. Would recommend adding morbidity, mortality data.

- Morbidity and mortality data are highlighted with blue in the introduction, and in the review's content, we mentioned a few elements when discussing different studies.

  1. A chart listing all the types of tests discussed and their characteristics would be summative.

- We did the requested chart – Table nr. 3

Reviewer 3 Report

The authors gave an overview of the PCR techniques used for diagnosing pneumonia, as well as their usefulness in the management of VAP, and made a comparison between the classic and bedside (point-of-care- POC) PCRs. Overall, the review is too lengthy and the content is not novel enough to reflect the recent progresses in the fields. Most of the references are outdated. At the same time, it is not well organized and the focus is not prominent. As a review, there should not be a Discussion part. Other minor questions;

1.      There is no necessary to cite reference in the abstract.

2.      References should be numbered in the order they appear.

3.      Line 58, “the left column: should be “the right column”?

4.      The nomenclature for the bacteria should be in italic. Line 60-63, and other places

5.      Line 99-101, 106, …  6 should be in superscript

6.      6. Line 106, it is not appropriate to use the word “germs”, “microorganism” or “bacteria” is more suitable

7.      Line 184-185, the full name of the abbreviation POCT should be indicated when it firs appears.

8.      Line 224, the name ply for a gene should be in italic.  

Author Response

Reviewer nr. 3

In the resubmitted manuscript, the modifications suggested for this reviewer are marked with the color green.

Comments and Suggestions for Authors

The authors gave an overview of the PCR techniques used for diagnosing pneumonia, as well as their usefulness in the management of VAP, and made a comparison between the classic and bedside (point-of-care- POC) PCRs. Overall, the review is too lengthy and the content is not novel enough to reflect the recent progresses in the fields. Most of the references are outdated. At the same time, it is not well organized and the focus is not prominent. As a review, there should not be a Discussion part.

  • We took almost two pages from the review with information that we considered to make the review too long but also which was not necessarily relevant to the primary purpose. Regarding the novelty of the review, we introduced more new studies in each section of the manuscript, especially the ones focusing on point-of-care PCR. In addition, we tried to reorganize the content to have a clearer focus. Finally, we took out the Discussion. All of the references were re-arranged with Zotero software and are in a proper order from the beginning to the end.

Other minor questions;

  1. There is no necessary to cite reference in the abstract.

- We took them out.

  1. References should be numbered in the order they appear.

- We modified the enumeration of the references to have what you requested

  1. Line 58, “the left column: should be “the right column”?

- We corrected this

  1. The nomenclature for the bacteria should be in italic. Line 60-63, and other places

- We modified this aspect throughout the whole manuscript.

  1. Line 99-101, 106, …  6 should be in superscript

- We took out this paragraph to shorten the manuscript, as instructed.  

  1. 6. Line 106, it is not appropriate to use the word “germs”, “microorganism” or “bacteria” is more suitable

- We changed the terms in the whole manuscript.

  1. Line 184-185, the full name of the abbreviation POCT should be indicated when it firs appears.

- We modified this and used only „POC” in the manuscript so that we wouldn’t generate any confusion.

  1. Line 224, the name ply for a gene should be in italic.  

- We corrected this.